# A Novel Germline Mutation of *ADA2* Gene in Two “Discordant” Homozygous Female Twins Affected by Adenosine Deaminase 2 Deficiency: Description of the Bone-Related Phenotype

**DOI:** 10.3390/ijms22158331

**Published:** 2021-08-03

**Authors:** Silvia Vai, Erika Marin, Roberta Cosso, Francesco Saettini, Sonia Bonanomi, Alessandro Cattoni, Iacopo Chiodini, Luca Persani, Alberto Falchetti

**Affiliations:** 1Department of Endocrine and Metabolic Diseases, IRCCS, Istituto Auxologico Italiano, 20145 Milan, Italy; s.vai@auxologico.it (S.V.); e.marin@auxologico.it (E.M.); i.chiodini@auxologico.it (I.C.); luca.persani@unimi.it (L.P.); 2IRCCS, Istituto Auxologico Italiano, San Giuseppe Hospital, 28824 Verbania, Italy; r.cosso@auxologico.it; 3Department of Pediatrics, Università degli Studi di Milano-Bicocca, Fondazione MBBM, San Gerardo Hospital, 20100 Monza, Italy; francesco.saettini@unimib.it (F.S.); sbonanomi@fondazionembbm.it (S.B.); alessandro.cattoni@unimib.it (A.C.); 4Department of Medical Biotechnologies and Translational Medicine, University of Milan, 20122 Milan, Italy

**Keywords:** rare diseases, DADA2 syndrome, *ADA2/CECR1* gene, bone metabolism, DXA analysis, bone health, neridronate

## Abstract

Adenosine Deaminase 2 Deficiency (DADA2) syndrome is a rare monogenic disorder prevalently linked to recessive inherited loss of function mutations in the *ADA2/CECR1* gene. It consists of an immune systemic disease including autoinflammatory vasculopathies, with a frequent onset at infancy/early childhood age. DADA2 syndrome encompasses pleiotropic manifestations such as stroke, systemic vasculitis, hematologic alterations, and immunodeficiency. Although skeletal abnormalities have been reported in patients with this disease, clear information about skeletal health, with appropriate biochemical-clinical characterization/management, its evolution over time and any appropriate clinical management is still insufficient. In this paper, after a general introduction shortly reviewing the pathophysiology of Ada2 enzymatic protein, its potential role in bone health, we describe a case study of two 27 year-old DADA2 monozygotic female twins exhibiting bone mineral density and bone turnover rate abnormalities over the years of their clinical follow-up.

## 1. Introduction

Adenosine Deaminase 2 Deficiency (DADA2) syndrome is a rare monogenic disorder, firstly described in 2014 (1) and mostly due to biallelic recessively inherited loss-of-function mutations in the *ADA2*/*CECR1* gene encoding for this enzyme. DADA2 is an immune system disease recognized as a mimic of polyarteritis polyarteritis nodosa (PAN) with systemic autoinflammatory vasculopathy, often presenting at infancy/early childhood, in fact caused by the absence of ADA2 enzymatic activity [1,2]. To date, more than 160 DADA2 cases have been reported in the literature [3]. Generally, DADA2 shows pleiotropic manifestations including stroke, systemic vasculitis (generally targeting skin, gut, and central nervous system), hematologic alterations, and immunodeficiency. Hematopoietic stem cells transplantation (HSCT)/bone marrow transplantation (BMT) represents the standard-of-care for those severe cases developing malignancies as forms of leukemia. As an extreme simplification, HSCT indicates transplanted stem cells, from the bloodstream, also referred to as peripheral blood stem cell transplant, and BMT indicates stem cells collected from bone marrow and transplanted into a patient. However, phenotypical heterogeneity, with a clinical wide spectrum, is also increasingly reported among family members sharing identical germline mutations. In fact, neither systemic inflammation nor vasculopathy spectrum has been uniformly described in DADA2 patients, and some of them may remain asymptomatic through adulthood. Although radiologically detectable bone defects have been described in approximately 50% of patients with early-onset of ADA deficiency, particularly anterior rib cupping, scapular spurring, growth failure as well as other skeletal abnormalities, detailed information on bone health is scarce in DADA2 syndrome. However, since similar bone alterations are also observed in other conditions of immunodeficiency, it is still undefined whether the bone abnormalities exhibited in DADA2 can be regarded as a result of the associated perturbations in purinergic metabolism or the effect of immunodeficiency itself [4]. As in other human skeletal pathological conditions characterized by the absence/reduced function of enzymatic activity, e.g., hypophosphatasia [5], an effective treatment should consist of the enzyme replacement therapy (ERT). Specifically, the described improvement of the DADA2-related skeletal disturbances reported in affected children after ERT, supports the hypothesis of a detrimental role of *ADA2* mutation upon bone health [6]. Here, after a general introduction shortly reviewing the pathophysiology of Ada2 enzymatic protein, its potential role in bone health including the interactions with specific bone cell receptors, we present a case study of two 27 year-old DADA2 homozygote female twins exhibiting bone mineral density (BMD) and bone turnover rate abnormalities over the years of their clinical follow-up.

### 1.1. General Information on Ada2 Enzyme

The Ada2 enzyme is ubiquitously expressed in human body, playing a pivotal role in many cellular processes. It is known to be expressed both at intracellular level and, in specific cell types, complexed with CD26 antigen on the cell surface [7]. It irreversibly catalyzes, as a part of the purine salvage pathway, the hydrolytic deamination of adenosine (Ad) and 2-deoxyadenosine (2-dAd) [8,9] to produce inosine (I) and deoxyinosine (dI) (Figure 1A,B), respectively, with a crucial role either in purine metabolism or Ad homeostasis by positively modulating Ad receptors (see below). Consequently, DADA2 determines the accumulation of either Ad or 2-dAd molecules, which cannot be further metabolized in Ada-deficient patients (Figure 1A,B). Large amounts of 2-dAd, a component of DNA, are generated from DNA degradation, as it occurs in cells/tissues with higher rate of apoptosis [7]. An uncontrolled increase in 2-dAd levels has been shown to be lymphotoxic either at the nucleoside level or after conversion to dATP by intracellular thymic deoxynucleoside kinase [10,11]. It has been reported that the amounts of residual Ada2 activity, rather than mutant genotype, may be more predictive of the resulting phenotype [12]. Recently, cases of lymphoproliferative disease with resemblance to large granular lymphocyte leukemia have been described in a Finnish cohort of DADA2 patients [13] and a lymphoproliferation, persistence of large granular lymphocytes, and T-cell perturbation have been reported in two siblings exhibiting novel *ADA2* variants, including the *P311L* variant (proline to leucine substitution) [14].

### 1.2. How Could the Role of Ada2 Protein in Bone Physiopathology Be Explained?

While considering the above described multiple actions of Ada enzyme in different cellular and tissue contexts, no direct evidence for its biological role in bone and mineral metabolism exists. Ada2 protein is expressed primarily by monocytes and macrophages [15,16], the latter known to be the precursors of osteoclast cell. However, osteoclast is not the only bone cell in which Ada activity can play a pathophysiological role, since osteoblasts also express it. In fact, an animal model study, with a threefold reduction in Ada activity in the mesenchymal progenitors of mature osteoblasts versus “healthy” mature osteoblasts expressing high levels of Ada2 enzymatic activity, indirectly suggests that an up-regulated Ada expression must occur for their physiological differentiation-maturation processes. Consequently, the Ada2 deficiency may play a potential severe detrimental effect on both the viability and function of osteoblasts. An in vitro experimental transfection approach, in which in Ada2-deficient osteoblasts a functional copy of *ADA* gene sufficiently corrected the related bone defects, supports this hypothesis [4]. The Receptor Activator of Nuclear Factor κB Ligand (RANKL), mainly produced by osteoblasts, and its decoy receptor osteoprotegerin (OPG) are the main cytokines regulating the physiological cross-talk between osteoblasts and OCLs. In particular, RANKL:OPG ratio modulates both the formation and activity of osteoclasts, clearly indicating these cytokines as important players in the physiological bone environment maintenance. Consequently, an adequate, appropriately controlled, space-time balance of functions and interactions between osteoblasts and osteoclasts is fundamental for maintaining bone homeostasis and, more generally, the skeletal health [4]. An equally balanced space-temporal relationship must also exist between production and local concentration of RANKL and OPG. Several considerations can be made regarding the potential role that may link the Ada2 enzymatic activity deficiency to skeletal damage. Interestingly, in addition to osteoblasts, activated T- and B-lymphocytes also produce RANKL, and, even more interesting, the bone marrow B-lymphocytes also express OPG. Since the Ada2 enzymatic activity deficiency associates with both T- and B-cell lymphopenia, this feature may suggest the occurrence of unbalancing of the RANKL-OPG axis equilibrium, partly attributable to an immune-dependent mechanism, favoring a final potential negative impact on both quantity (mass) and quality of the bone. In an Ada knock-out mice model, a reduction in osteoblastogenesis has been demonstrated to be associated with low bone formation [4]. Interestingly, consistently with these findings, an abnormal bone clinical phenotype has been recently described in those children undergoing ketogenic diet therapy who developed a ketotic hypercalcemia, highly likely depending by the fact that ketogenic diet reduces Ad kinase activity, consequently both prolonging Ad serum half-life and increasing interaction with Ad receptor, without impairing the osteoclast differentiation [17]. This body of evidences is indirectly suggesting that the hyperaccumulation of Ad and 2-dAd, linked to the Ada2 deficiency, may contribute to alter homeostasis and bone health.

### 1.3. Ada2 Receptors in Bone Cells: Co-Starring Actors

Either osteoblasts or osteoclasts express Ad receptors, a group of G protein-coupled receptors (GPCR). Consequently, an abnormal Ad signaling, following the increased Ad concentrations, may play an important role in modulating their activity and be a causal factor in the development of bone abnormalities associated with DADA2 [4]. Currently, four subtypes of Ad GPCRs are known to be expressed by osteoblasts and osteoclasts (A1, A2A, A2B, and A3), and conflicting effects have been described [18]. At osteoclast level, A1 receptors have an important role for their differentiation while at osteoblast level the stimulation of A1 favors the decrease in serum alkaline phosphatase (ALP) activity, a bone formation marker. More interestingly, all the other Ad receptors seem to play an opposite effect, promoting the bone formation activity by osteoblast and inhibiting the osteoclastogenesis [19]. In addition, subjects affected by severe combined immunodeficiency (SCID) due to mutations in the *ADA* gene, showed impairment of both significant osteoblasts and bone formation [20]. All these findings indirectly support that the Ada2 deficiency-related bony alterations may also occur as a non-immunological manifestation, not entirely explainable as related to the observed immunodeficiency. In an Ada-deficient mice model, ERT, HSCT/BMT, and gene therapy (GT) determined near-complete corrections of the bone-related abnormalities, as also happens in some ADA2 deficient patients, even if this effective response has not been fully replicated in other cases, also due to differences existing in the experimental protocol applied to mice model [4]. Therefore, when a full understanding of the immune-dependent and immune-independent pathogenesis will be achieved, this might lead to improve therapies and correction of bone abnormalities associated to Ada2-deficient activity.

## 2. Case Reports

DADA2 homozygote female twins, II-1 Tw1 and II-2 Tw2 (Figure 2) were initially followed for neutropenia, then, due to vasculitis events. Clinical investigations were performed for the suspected DADA2 confirmed in February 2017, at the age of 23 years. Table 1 shows the chronological occurrence of bone-unrelated events for both the twins. Remarkably, II-2 Tw2 developed large granular lymphocytes Leukemia (LGLL), while II-1 Tw1 has not developed any oncological disorder for the time being.

The complete family pedigree, identified as ADA2-AUXMI1, is reported in Figure 2. Interestingly, both parents (I-1 and I-2) did not show any biochemical-clinical abnormal phenotype involving bone disorders nor systemic autoimmune disease, except for Raynaud’s phenomenon exhibited by the mother.

Both twins followed a free Western Mediterranean type diet, with a daily calcium intake of about 750–800 mg. Such a diet is still currently followed. II-1 Tw 1 referred menarche at 13 years of age and she is currently having regular menses, whereas II-2 Tw2 referred menarche at 12 years of age, with irregularity of the following cycles. As also specified later in the text, after undergoing HSCT, II-2 Tw2 presented with secondary amenorrhea and initiated appropriate oral estrogen-progestogen therapy. Regular physical activity, such as free-body gymnastics, at least two times a week, has been always carried out by both twins. Interestingly, they have a discordant phenotypical expressiveness, as also reported in other rare bone diseases such as X-linked-hypophosphatemic rickets [21]. All the clinical-instrumental and biochemical parameters to investigate bone health in our cases are here specifically reported. 

## 3. Materials and Methods

These patients underwent an extensive assessment of bone health at the Auxologico Italiano, Milan, Italy, according to common clinical work-up procedures, as indicated by the Quality Office of the Internal Health Management Board of the Institute (last internal revision 3 July 2018), and according to the Italian GCPs released by the Italian Ministry of Health. Information was collected, as detailed as possible, aimed at ascertaining/excluding possible consanguinity of the parents, going back to the great-grandparents, from which no data supporting the consanguinity of the parents emerged, although this procedure does not allow it to be absolutely excluded. Both twins gave their informed consent, as an internal policy of the institution approved by the Internal Board, for the treatment of personal data, for the clinical-biochemical, genetic, ionizing radiation, procedures and neridronate treatment (II-2 Tw2). Signed informed consent was also obtained from the parents for the genetic analysis and data management.

### 3.1. Bone Phenotyping

#### 3.1.1. Dual Energy X-ray Absorptiometry (DXA)

The bone mass was monitored by DXA-BMD scans at lumbar spine (LS), L1-L4, total hip/femoral neck (TH/FN) and total body (TB) but head, (TBLH), through 2001–2020 (Figure 3 and Figure 4A–C). Hologic Discovery A scanner was used before 2016, while Hologic Horizon A scanner (Hologic Inc., Bedford, MA, USA) was used after 2016. DXA results were presented as absolute values, g/cm^2^, T-scores (the number of standard deviations (SDs) that a patient’s BMD differs from that of a healthy, sex-matched reference population at their peak bone mass) and Z-scores (the number of SDs by which a patient’s BMD differs from that of a healthy ethnicity, age- and sex-matched control population). According to the Official position 2019 of the International Society for Clinical Densitometry (ISCD), for the BMD reporting in females prior to menopause and in males younger than age 50, Z-scores, and not T-scores, have to be preferred (https://iscd.org/, 28 May 2019). TB % fat composition results are described in Figure 5A,B.

#### 3.1.2. Spine X-rays

The patients underwent anterior-posterior (AP) and latero-lateral (LL) X-rays of dorsal (D), and LS to investigate the presence of vertebral deformities/abnormalities in 2011, 2018, 2019, and 2020 (not shown).

#### 3.1.3. Bone-Mineral Metabolism, and Hormonal Biochemical Analyses

Bone metabolism parameters through 2011–2020 are reported at Figure 6 (A-A1 for IL-1 Tw1 and B-B1 for IL-2 Tw2). Blood and urine samples were collected in the morning following an overnight fasting, and after an appropriate 24-h collection, respectively. The following analyses were performed to assess either bone metabolism or bone turnover markers (BTMs): serum and urinary calcium (s-Ca and 24hU-Ca), phosphate (s-PO and 24hU-PO), magnesium (s-Mg and 24hU-Mg), and serum alkaline phosphatase (s-ALP) were measured with standard colorimetric laboratory methods; serum bone-specific alkaline phosphatase (s-BSAP) was measured by immune-enzymatic assay (EIA, Technogenetics, Milano, Italy); serum parathyroid hormone (PTH), C-terminal telopeptide (CTX), plasma osteocalcin (P-OC), urinary N-terminal telopeptide (U-NTX), 25-OH vitamin D, thyroid-stimulating hormone (TSH), adrenocorticotropic hormone (ACTH), luteinizing hormone (LH), follicle-stimulating hormone (FSH), estradiol, free triiodothyronine (FT3) and thyroxin (FT4) were measured by means of electrochemiluminescence immunoassay (ECLIA, Roche Diagnostic, GMBH, Mannheim, Germany). Finally, 1,25OH vitamin D was measured by means of chemiluminescence (CLIA, Immunodiagnostic Systems, Tyne and Wear, UK). Mineral and bone metabolism evaluation was assessed yearly.

### 3.2. Mutational Analysis

The DNA germline mutational analysis has been firstly performed in proband II-2 Tw2, December 2016, and then, once the *ADA2* gene germline mutation had been identified, it was extended to the other family members. Mutational analysis procedures followed the Sanger’s approach, as widely reported in literature [22].

### 3.3. ADA2 Enzymatic Activity

ADA2 plasma levels were detected with ELISA kit (eBioscience, Thermo Fisher Scientific, Waltham, Massachusetts, US). ADA2 activity was assessed in primary monocytes. Peripheral bone marrow mesenchymal cells (PBMCs) were isolated through Ficoll-Paque and monocytes isolated by adherence, incubated for 1 h in 24-well plate with RPMI 1% penicillin–streptomycin (Sigma Aldrich Italia, Milan, Italy), 1% l-glutamine (Euro-Clone) in 5% CO_2_ at 37 °C. Monocytes were then cultured in PBS in the presence of exogenous adenosine (Sigma Aldrich Italia, Milan, Italy), with or without the ADA1 inhibitor (erythro-9-(2-hydroxy-3-nonyl)adenine, EHNA (Sigma Aldrich Italia, Milan, Italy). After 4 h of incubation at 37 °C and 5% of CO_2_, supernatants were collected and the activity evaluated through the measurement of the adenosine-derived products (inosine, hypoxanthine) in high-performance liquid chromatography (HPLC). Results were compared with 24 healthy controls.

## 4. Results

### 4.1. DXA Scans

#### 4.1.1. II-1 Tw1

DXA scans revealed LS-BMD values significantly lower than sex- and age-matched subjects from Caucasian Italian ethnical group, according to the International Society for Clinical Densitometry (ISCD) [23], with Z-score lower than -2.0 SD: −2.2 SD (2015), −2.4 SD (2017), −2.6 SD (2019), and −2.4 SD (2020) (Figure 3A,B,D). On the other hand, the TH/FN DXA scans exhibited Z-score values significantly higher than −2.0 SD, indirectly suggesting a reliable major sufferance of the skeletal sites with a prevalent trabecular component (Figure 3A, B, D) [24]. Sequential TB DXA scans, from 2011 up to 2020, also suggested a significant lower BMD, with Z-scores of −2.1 SD (2011), −2.2 SD (2013), −2.5 SD (2017), −3.0 SD (2019) (not shown), and −3.1 SD (2020) (Figure 3C), respectively. Interestingly, the general worsening of the TB Z-score from 2013, −2.2 SD, to 2017, −2.5 SD, coincides with the reduction in the percentage of fat, always evaluated by TB DXA scans, 31.4% in 2013 and 24.8% in 2017 (Figure 4A). However, contrary to what has now been described, the progressive reduction in the skeletal TB Z-score after 2017 is accompanied by an increase in the percentage of TB fat tissue in 2019, 35.1% compared to the previous 24.8% in 2017, thus stabilizing in 2020, 34.3% (Figure 5(A,A1)), in the presence of a stable weight of 47 Kg (Figure 5(A1)).

#### 4.1.2. II-2 Tw2

She also exhibited a significantly reduced BMD (Figure 4A,B), with Z-score values at both LS and FN level lower than -2.0 SD. The follow-up DXA scans showed a worsening trend in Z-score values at both skeletal sites, except for 2019 with a countertrend Z-score value of −2.3 SD, up to a clear improvement in 2020 LS-BMD, Z-score of −1.5 DS (Figure 4D). Differently than II-1 Tw1, II-2 Tw2 FN Z-scores were always associated to a significantly reduced BMD (Figure 4D). Figure 4C shows the 2020 TB scan. The box under the image reports the BMD results from 2011 up to 2020 with a significant scan by scan reduction starting from 2013. In particular, the observed TB Z-score reductions from 2013 to 2019, −2.1 SD, −2.8 SD, −3.6 SD, and −3.2 SD, respectively, associated with varying TB percentage fat values, 35.4%, 27.6%, 46.1 SD, and 42.7% (Figure 4B) with body weight showing minimal, not significant, fluctuations between 48 and 53 Kg (Figure 5(B1), gray boxes).

### 4.2. Spine X-rays

#### 4.2.1. II-1 Tw1

D-L, AP and LL, spine X rays have been performed in 2011, 2019, and 2020 (not shown) exhibiting “D-L italic S-inflection, right-convex in the dorsal site and without consensual rotation of the metamers”. In 2011 and 2019, the radiological report was identical as “the physiological kyphosis on the LL plane is preserved. Osteo-structural failures of the considered dorsal metamers are lacking, with inter-somatic disk spaces preserved. D-L italic S-shaped inflection, left-convex in the lumbar region and with slight hint of consensual rotation of metamers. Preserved the physiological lordosis on the LL plane”. On the 2020 control, the report slightly changed into “minimal scoliosis with opposing curves of the D-L spine. D-L metamers in the sagittal plane. Minimal spondylitis notes of the dorsal metamers at the kyphotic fulcrum. The dorsal interbody spaces are substantially preserved. No appreciable significant alterations of the lumbar metamers; lumbar interbody spaces preserved”.

#### 4.2.2. II-2 Tw2

D-L, AP and LL, spine X rays have been performed in 2011, 2019, and 2020 (not shown). In 2011, the DLS X-rays revealed “D-L-S spine in axis on the AP plane, with preserved physiological lordosis on the LL plane. spine. Widespread osteopenic notes, without the relief of osteo-structural subsidence of the vertebral bodies considered. Preserved the inter-somatic disk spaces”. In 2019, DLS X-rays confirmed/evidenced “slight Italic D-L S-shaped inflection on the AP plane, with preserved kyphosis and lordosis on the L-L plane. No sagging of the D-L vertebral somas considered are appreciated. The inter-somatic disk spaces are within the limits”. On the 2020 control, the previous results were reconfirmed.

### 4.3. Biochemical Analyses

The bone- and mineral-related biochemical results of II-1 Tw1 and II-2 Tw2 are summarized in Table 2. Although with some temporal discontinuities, these surveys cover a period between 2011, 18 years of age, and 2020, 27 years of age. Hormonal tests, not described, were all within the normal ranges, but gonadotropin values in II-2 Tw2, abnormal as for secondary amenorrhea occurred post-HSCT (2018). Routine biochemical tests, including uricemia, were all within their own reference range.

#### 4.3.1. II-1 Tw1

Differently from II-2 Tw2, II-1 Tw1 did not suffer from changes in immune-hematological components. Regarding BTMs, she exhibited accelerated bone turnover rate, with increases in both the bone formation marker P-OC, in 2011, 2013, 2019 (47, 44, and 49 mg/L, respectively) and the bone resorption markers CTX, in 2011, 2013, 2015 (659, 739, and 607 ng/mL, respectively) and U-NTX (56 BCE/mmol creatinine) in 2020 (Table 2 and Figure 6). The s-BSAP (Table 2 and Figure 6) was always within the normal range interval, as also were the levels of s-Ca/24U-Ca, s-PO/24U-PO, and s-Mg/24U-Mg (Table 2. Serum 25OHD circulating levels were below 30 ng/mL, the cut off limit of adequacy recommended in Italy to maintain a good bone health [25], in two tests, October 2011 (13 ng/mL) and March 2015 (22.4 ng/mL) (Table 2, although the suggested cholecalciferol supplementation was reported to have always been taken regularly. However, the serum 1,25OH vitamin D (not reported) and PTH values were stable within the normal range. In summary, an increased bone turnover, both P-OC and CTX higher than the upper limit of normal range, were described in 2011 and 2013 (Table 2 and Figure 6), and in 2011 an insufficient serum 25OHD was also observed. CTX and slightly reduced 25OHD serum levels were associated also in March 2015, while P-OC only increased in 2019 (Table 2 and Figure 6).

#### 4.3.2. II-2 Tw2

Regarding BTMs, she exhibited an increase in P-OC in 2018 (192 mg/L) and 2019 (104 mg/L) and increase in the bone resorption markers CTX and U-NTX. Specifically, in 2011, 2018 for CTX and 2011, 2019, and 2020 for U-NTX (Table 2, Figure 6). The s-BSAP values were slightly higher than the upper limit of referring range on December 2018 (40.1 mg/L), and October 2019 (33.2 mg/L) (Table 2 and Figure 6). Serum Calcium (S-Ca), and s-Mg levels were within the referring range whereas s-PO was slightly higher than the upper limit of range (5.1 mg/dL) in October 2019 (Table 2. Serum 25OHD circulating levels were insufficient in October 2011 (14 ng/mL) to then stabilize within the normal range in all subsequent checks (Table 2, under cholecalciferol supplementation. The serum 1,25OH vitamin D (not reported) was stable within the normal range, whereas PTH values were increased at the first test in October 2011 (72.9 ng/L) (Table 2, concomitantly to lower serum 25OHD levels (Table 2), as also in June 2019 and 2020 tests (86.2, 76.2, and 76.8 ng/L, respectively) (Table 2). All the 24-h urinary parameters were within the referring range values in all the performed tests (Table 2). In summary, in November 2011, vitamin D insufficiency associated with higher PTH (Table 2) and CTX values (Table 2 and Figure 6), and, in December 2018, all the BTMs (s-BSAP, P-OC, s-CTX, and U-NTX) values were increased (Table 2 and Figure 6). In June 2019, PTH, P-OC, and U-NTX were higher than upper normal limits (Table 2), while in October 2019, s-BSAP, s-PO resulted, and U-NTX was slightly increased (Table 2 and Figure 6). Finally, s-PTH, in the 2020 tests, was increased despite normal values of 25OHD, s-Ca/24U-Ca, and s-PO/24U-PO (Table 2). Table 3 summarizes the current pharmacological therapy of II-1 Tw1 and II-2 Tw2.

### 4.4. Mutational Analysis

The Sangers’ sequencing DNA test revealed a novel de novo *c.T203C:pL68P* heterozygous germline mutation at exon 3 of *ADA2*/*CECR1* gene (Figure 7) in both twins, while both the parents, I-1 and I-2 (Figure 1), resulted not to be carrier of this mutation. The variant *L68P* corresponds to variant *L188P* in the A isoform of the protein and it was found in 2 alleles out of 282248 alleles analyzed, with 0 homozygosity and an allelic frequency equal to 0.000007086 (ClinVar: https://www.ncbi.nlm.nih.gov/clinvar/variation/421491/#id_first last accessed on 29 January 2019; gnomAD: https://gnomad.broadinstitute.org/variant/rs760102576?dataset=gnomad_r2_1 last accessed on 7 July 2021).

### 4.5. ADA2 Functional Analysis

The ADA2 activity functional test (not shown) resulted as completely absent, compatible with failure in producing both inosine and d-inosine metabolites (Figure 1A,B), confirming the diagnosis of ADA2 deficit.

## 5. Discussion

The de novo *L68P* germline variant found, at exon 3 of *ADA2*/*CECR1* gene, is located in the dimerization domain of Ada2 protein and consists of a leucine (L) to proline (P) substitution or of a substitution between two hydrophobic amino acids. It is well known that changes of a single base, in the first position of the codon, usually give rise to a different amino acid, even if sharing similar chemical-physical properties. The P lateral group consists of a ring that does not fit into an ordered secondary structure, tending to interrupt the linearity of the polypeptide. This substitution occurs at a position that is conserved across species. As mentioned above, the *L68P* variant corresponds to variant *L188P* in the A isoform of the protein that has not been previously reported as a clear pathogenic variant, nor as a benign variant. However, we did not observe this variant in approximately 6500 individuals of European and African American ancestry in the NHLBI Exome Sequencing Project, indicating it as not a common benign variant in these populations. Thus, this variant, a semi-conservative amino acid substitution, may impact secondary protein structure as these residues differ in some properties. A previously published in-silico analysis predicted A isoform *L118P* mutation, affecting the catalytic domain, probably to damage the protein structure/function [26], and, consequently, it has been regarded as a variant of uncertain significance (https://www.ncbi.nlm.nih.gov/clinvar/variation/421491 last accessed on 29 January 2019). In the literature, the *ADA2* gene mutation, associated with DADA2 phenotypes, are most frequently reported to be in a homozygous state, biallelic inactivating mutations [3]. We identified a heterozygous germline mutation, but this may be since we performed the genetic investigation using the Sanger’s method, and a possible condition of compound heterozygosis cannot be definitively excluded. In fact, with this method, mutations such as insertions/deletions (indels) greater than 300 base pairs (bps) of DNA in the wild type allele may be missed, as may also be any mutations in another unsuspected gene. Moreover, medium indels, 50–300 bps, may also escape the identification. Moreover, according to their location within gene organization, intronic and promoter mutations may be not easily identifiable either [27]. Finally, we cannot exclude the hypothesis that *L68P/L118P* mutation may determine the silencing of the protein transcription of the “healthy” allele, through unknown mechanisms. The latter hypothesis is unlikely and difficult to sustain. The null enzyme Ada2 activity, lacking any type of specific residual Ada2 enzymatic activity, in both the twins suggested that the “wild type” allele is possibly involved, but if in a regulatory region or splice zone not covered by the primers, it is unknown. However, as reported in the literature, the amounts of residual ADA2 activity may be more predictive of phenotype rather than the resultant genotype [12].

Highly variable clinical findings among family members sharing the same homozygous *ADA2* mutations, as well as in homozygous twins, have been reported [2,3,12,28]. Interestingly, both II-1 Tw1 and II-2 Tw2 shared similar anthropometric features, nutrition habits, physical activity, and lifestyles, therefore the environmental component relating to these aspects can be regarded as homogeneous and should not have played a fundamental pathogenetic role. It is known that several risk factors for bone loss following HSCT exist. They include preparative therapeutical strategies in the setting of HSCT, induction of a premature menopausal status, and drugs to prevent and treat acute and chronic graft versus-host disease (GVHD). Then, most women become menopausal subsequent to HSCT/BMT. Moreover, younger individuals undergoing HSCT/BMT may have an impaired achievement of peak bone mass favoring a lately increased fracture risk [29]. These aspects could partly account for the different bone aggressiveness in our twins, with II-1 Tw1 having so far, a relatively less aggressive behavior than II-2 Tw2 (Figure 1, Table 1A,B). 

Specifically, II-1 Tw1 showed, since 2015, an unexpected low LS-BMD (Figure 2) and vitamin D insufficiency, lately supplemented. The DXA-BMD findings of II-2 Tw2 were certainly worse than II-1 Tw1, involving both a predominant trabecular structure (LS) and a predominant bone cortical structure (TH) (Figure 3D), furtherly worsened, particularly after HSCT in 2018 (LS-BMD = 0.687 g/cm^2^, Zs −3.5 SD, TH-BMD = 0.608 g/cm^2^, Zs −3.1 SD) (Figure 3A,B,D). Both twins showed increased BTMs levels, but while for II-1 Tw1 this was prevalent in the period 2011–2015, at 17–21 years of age (Table 2, Figure 6), for II-2 Tw2 such an increase was particularly prevalent after HSCT (Table 2, Figure 6) associated with the pharmacotherapy to prevent/treat GVHD (Table 1B). The most studied drugs for the prevention of HSCT-induced bone loss are amino-bisphosphonates (a-BPs), pamidronate or zoledronic acid [30,31], also able to increase bone mass [32,33]. In Italy, the drug regulatory agency (AIFA) has approved the therapeutic use of the a-BP neridronate as an anti-fracture drug in Osteogenesis Imperfecta from pediatric age [34]. Positive results with neridronate have also been reported in the literature in a context of bone dysplasia [35]. II-2 Tw2, at young-adult age, received infusions with i.v. 100 mg of neridronate every 2 months until the end of 2020, of course together with adequate vitamin D supplementation. This therapeutical approach resulted, in 2020, in a significant increase in BMD at both LS (0.908 g/cm^2^, Zs −1.5 SD), and TH (0.718 g/cm^2^, Zs −2.1 SD) sites, together improving the bone turnover rate, except in U-NTX (Table 2, Figure 6). Regarding the latter, these data appear to be in contrast with the positive results obtained with DXA scans. In fact, the use of U-NTX has been suggested to monitor response to BPs in the treatment of osteoporotic subjects for an early identification of non-compliance or presence of secondary osteoporosis. In particular, a greater decrease in U-NTX at 4 months has been reported to be associated with a greater increase in spine BMD at 18 months of therapy [36]. In the case of II-2 Tw2, an elevated value of U-NTX was already present in 2011, at the age of 17, persisting higher also in 2019 and 2020.

## 6. Limitations and Conclusions

Certainly, in this specific case we cannot be sure that the persistence of increased U-NTX values can be really interpreted as a lack of response to a-BP therapy, since in 2019 and 2020 they could also be influenced by HSCT (2018) and therapies to prevent GVHD (Table 1B). Of course, in II-2 Tw2, the subsequent appearance of premature ovarian failure (POF) may also have strongly contributed to the greater severity of the bone mass phenotype than II-1 Tw1 who, even if not suffering from HSCT nor POF, exhibited a reduced BMD too. The significantly lower BMD values on both twins may be due to a failure in achieving adequate peak bone mass, but whether this directly links to DADA2 syndrome, and/or to a general genetic make-up and/or unrecognized environmental factors and/or concomitance polytherapy is still uncertain. Of course, both the twins have not reached adequate BMD values over the years. Certainly, the important phenotypic discordance in these homozygous twins, particularly regarding the oncological component, suggests several hypotheses, which are not mutually exclusive: (1) the need to deepen the genetic analysis to ascertain/exclude a still unidentified mutation in the wild allele of the *ADA2* gene; (2) the roles of epigenetic mechanisms or modifying genes contributing to low BMD, other than to the discordant oncological phenotype; and (3) the possibility that stochastic event(s), difficult to be identified, have played or are playing a role in such phenotype discordance. Up to now, the X-ray monitoring of the spine has not shown alterations in the vertebral morphometry, excluding, now, alterations attributable to bone fragility, although we cannot exclude that skeletal fragility may manifest itself at the level of other bone districts/segments in the future. Certainly, a radiological follow-up, at least at spine level, perhaps more rarefied over time, should always be considered, together with a periodical evaluation of BTMs, calcium-phosphate metabolism, and more general biochemical aspects. Of course, II-2 Tw2 continues in her periodic oncologic follow-up, but attention to hematological parameters must also be suggested to II-1 Tw1 who has not shown any anomalies in this sense, so far. In conclusion, our preliminary data confirm that in DADA2 patients adequate/appropriate bone health status should be monitored together with all those parameters linked to the clinical manifestations more frequently represented in the phenotypical spectrum of this syndrome. Moreover, since II-1 Tw1 and II-2 Tw2 originally exhibited neutropenia, when this abnormality is present in young subjects, not explainable in another way, evaluation of bone health could be considered. Of course, when a severe, significant, loss of BMD is observed in DADA2 patients, especially concomitantly with oncological disorders and/or conditions compromising the skeletal health, an appropriate anti-fracture therapy, combined, when needed, with adequate supplementation/supplementation of calcium and/or vitamin D should be advised and considered. Even if more consistent clinical data are lacking, it could not be a priori excluded that in subjects inappropriately osteoporotic/osteopenic for their age, regardless of the presence/absence of features belonging to the clinical spectrum of DADA2 syndrome, a “hide” heterozygous *ADA2* genetic alteration, and/or a shading underlying abnormal purinergic metabolism may exist. Finally, encouraging preliminary observations are coming from studies using ADA2-ERT suggesting improvements of hepatocellular abnormalities [37,38], pulmonary alveolar proteinosis [39,40], and bone dysplastic features [6], all associated with untreated ADA deficiency.

## Figures and Tables

**Figure 1 ijms-22-08331-f001:**
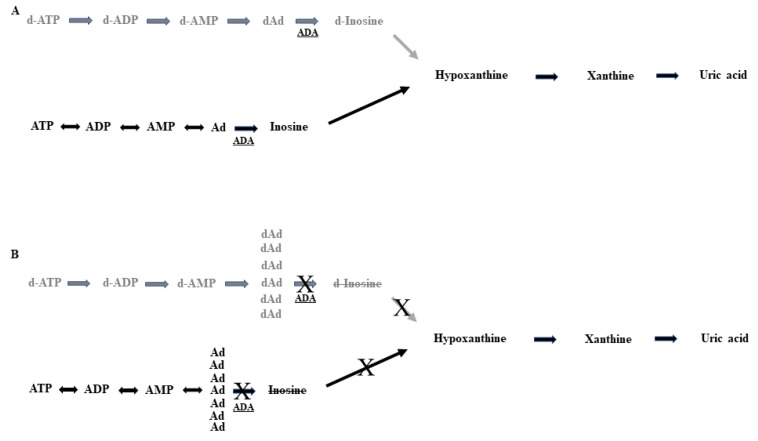
(**A**) Normal integral metabolic purine salvage pathway enables the regulation and availability of purines; (**B**) Disruption of ADA activity prevents the physiological continuation of the purine pathways with increased levels of Ad and dAd. In grey letters, the deoxy-metabolites branches are reported. ADA2 inactivation determines the stop of the dAD to d-Inosine and Ad to Inosine progression, and consequently the downstream final product represented by uric acid, with upstream accumulation of both dAd and Ad metabolites.

**Figure 2 ijms-22-08331-f002:**
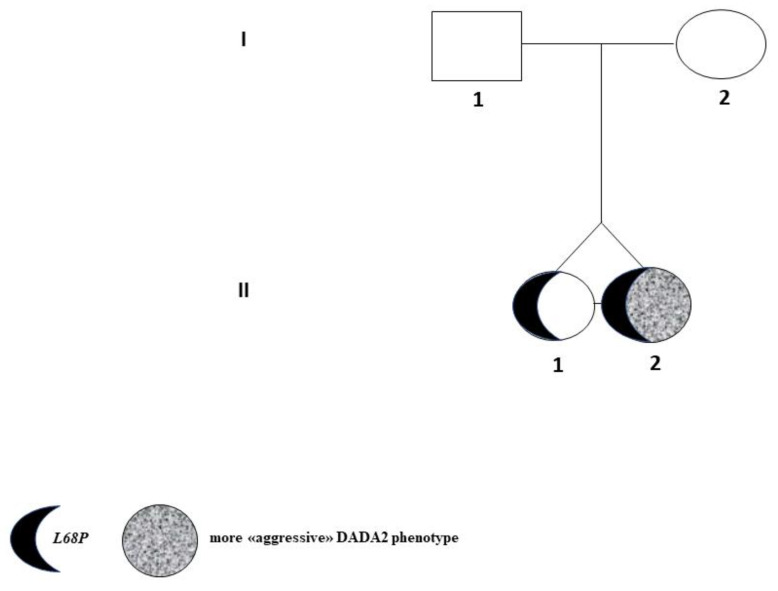
Pedigree of family ADA2-AUXMI1.

**Figure 3 ijms-22-08331-f003:**
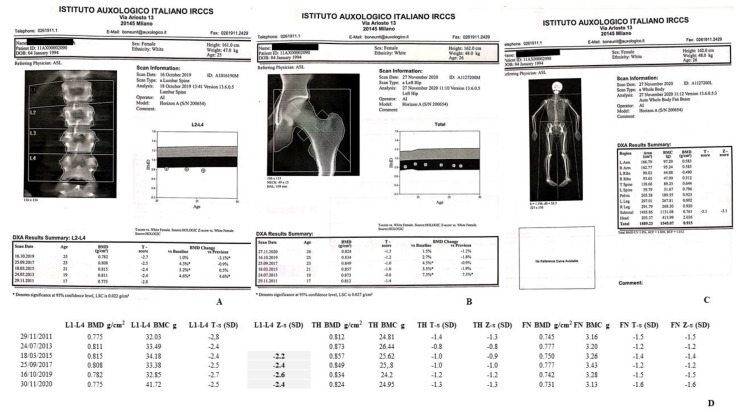
(**A**–**D**). II-1 Tw1 LS- (A), TH/FN- (**B**), and T. B.-DXA (**C**) scans. A e B report also the previous relative findings, from 2011 to 2019, showing the trend of BMD over the years. Asterisk refer to values that are significantly different from the previous ones (result calculated directly by the software of the device). (**D**) II-1 Tw1. It reports also LS, TH and FN Z-score. In bold are shown the Z-scores (LS) which, in accordance with ISCD, diagnosed the reduced bone mass compared.

**Figure 4 ijms-22-08331-f004:**
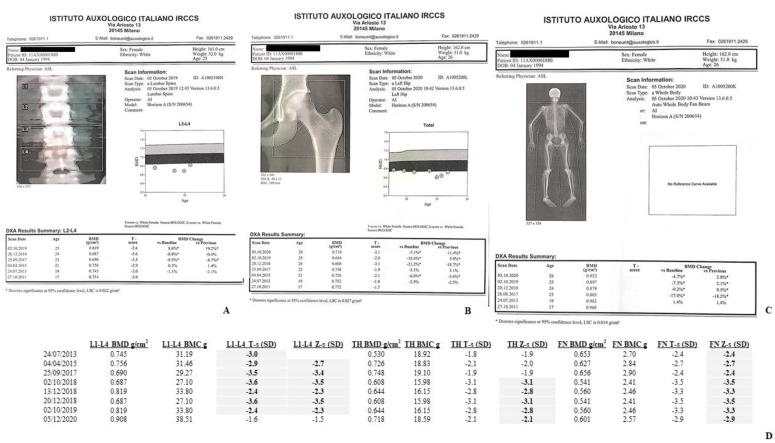
(**A**–**D**). II-2 Tw2 LS- (**A**), TH/FN- (**B**), and T. B.-DXA (**C**) scans. A e B report also the previous relative findings, from 2011 to 2019, showing the trend of BMD over the years. Asterisk refer to values that are significantly different from the previous ones (result calculated directly by the software of the device). (**D**) II-2 Tw2. It reports also LS, TH and FN Z-score. In bold are shown the Z-scores (LS) which, in accordance with ISCD, diagnosed the reduced bone mass compared.

**Figure 5 ijms-22-08331-f005:**
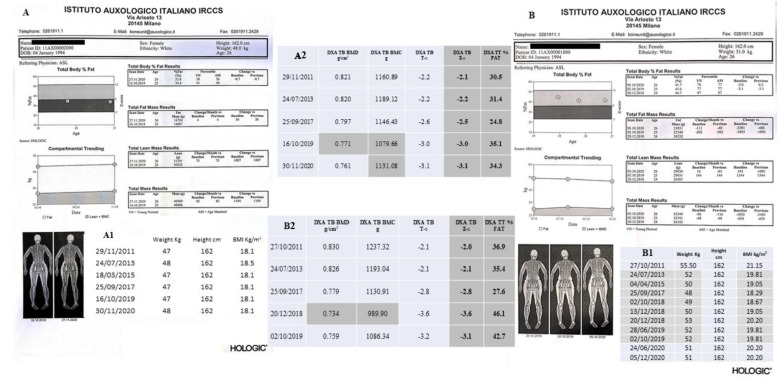
(**A**) II-1 Tw1 TB% fat composition results, 2019 vs. 2020. (**A1**) reports anthropometric results from 2011 up to 2020, and (**A2**) summarizes DXA TB parameters (BMD, BMC, T- and Z-scores, and total fat percentage) from 2011 up to 2020. (**B**) II-2 Tw2 TB% fat composition results, 2018, 2019 and 2020. (**B1**) reports anthropometric results from 2011 up to 2020, and (**B2**) summarizes DXA TB parameters (BMD, BMC, T- and Z-scores, and total fat percentage) from 2011 up to 2019.

**Figure 6 ijms-22-08331-f006:**
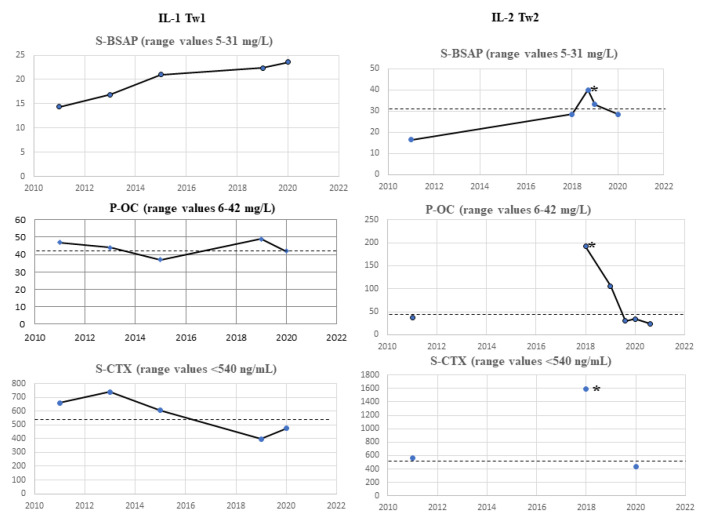
BTMs trend over time (U-NTX is not described here, which is instead described in Table 2). The dashed line in the graphs represents the upper limit of the range for each parameter. On the right side of each figure are the respective units of measurement, indicated in the parenthesis next to each BTM. The line below each graph shows the year in which the relevant biochemical assessments were carried out. The * symbol indicates the start of amino-bisphosphonate therapy (stopped at the end of 2020).

**Figure 7 ijms-22-08331-f007:**
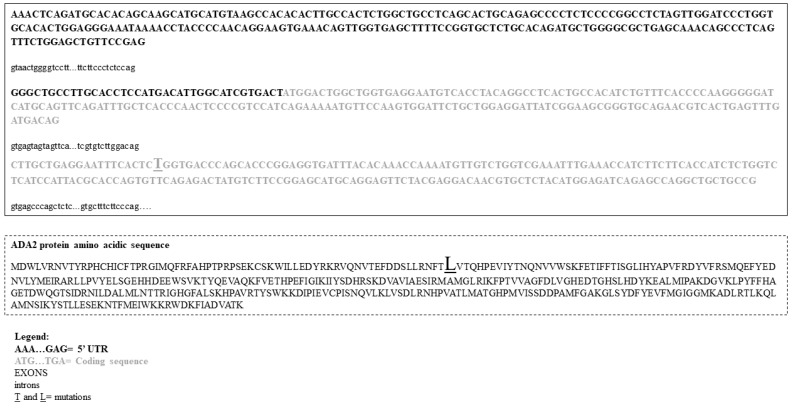
Within the solid box, a partial DNA sequence of *ADA2* gene is reported, while in the dashed box the corresponding amino acid sequence of ADA2 protein is depicted. The larger and underlined letters indicate the gene mutation (T) and amino acid substitution in the corresponding protein (L), respectively.

**Table 1 ijms-22-08331-t001:** Chronological order of bone unrelated abnormalities occurrence [(**A**): II-1 Tw1. (**B**): II-2 Tw2]. (**B**) contains also information on therapies performed by II-2 Tw2 in 2016–2018-time interval.

**(A) II-1 Tw1.**
1996 and 1997. She had the occurrence of two ischemic strokes.
2016. She received the diagnosis of autoimmune primary hypothyroidism.
**(B) II-2 Tw2.**
1996. At 3 years of age, she developed a CD19 deficiency (51 mm^3^, normal range values 200–2100 mm^3^) with hypogammaglobulinemia.
2007. Occurrence of episodes of acrocyanosis and vasculitis phenomena in the lower limbs.
2008. On January, hematological findings of leukopenia and neutropenia. Left knee pain, radiating to the leg (anteriorly). Knee X-ray performed on 25.08 was negative. The orthopedic visit indicated the need for analgesic therapy only. Subsequently, she referred the onset of burning pain and purplish erythematous lesions only partially vesicle/crusted, located only in the left limb, calf, tibial region and forefoot.
2009. For suspected vasculitis, a skin biopsy was performed and the findings were compatible with leukocytoclastic vasculitis.
2010. On December, immunological investigation revealed positivity of anticardiolipin and antiphospholipid antibodies.
2011. A granulocyte colony-stimulating factor (G-CSF)-based treatment started from 6 April with benefit.
2012. On December, she had cryopreservation of oocytes.
2016–2017. On October 2016, she started therapy with hydroxychloroquine sulfate 1 tab/day, continued until June 2017.
July 2017: she started steroid therapy, prednisone at 2 mg/kg/day (firstly per i.v. and then per os), prolonged up to September 2017, when it was reduced in a scalar way together with the start of Etanercept (Enbrel) therapy. July–September 2017: evidence of large granular lymphocytes (LGL) aspect with associated immunophenotype of 55.3% CD57^+^ and T Lymphocytes-Large Granular (TLLG) aspect, suggesting the onset of LGL Leukemia (LGLL). Such a suspicion was then confirmed by bone marrow analysis and the presence of T cell receptor (TCR) α (TCR α) (about 45%) and TCRβ (approximately 35%) oligoclonality, together with neutropenia, associated with episodes of vasculitis in the lower limbs, refractory to G-CSF and occurrence of steroid dependence.
2018. On 15 February, she underwent HSCT. The source of transplant was represented by bone marrow stem cells (BMSCs) from matched unrelated donor (MUD). A Treosulfan-Thiotepa-Fludarabine (Thio-Treo-Flu)-based conditioning regimen treatment was performed prior than allogenic transplantation. The graft consisted of 2.45 ×10^8^ total nucleated cells (TNC)/kg, 3.02 × 10^6^ CD34^+^/kg, 48.34 × 10^6^ CD3^+^/kg. The HLA matching resulted to be 9/10 in ratio. The acute graft versus host disease (GVHD) prophylaxis consisted of Rituximab, Methotrexate, and Cyclosporine. Cryopreserved hematopoietic stem cells (HSCs) of the allogeneic donor have been made available. Then, secondary amenorrhea, induced by HSCT and related treatment, occurred. In April, she suffered for the appearance of neuropathic pain for which she started oral gabapentin therapy followed by important symptomatologic improvement.

**Table 2 ijms-22-08331-t002:** II-1 Tw1 and II-2 Tw2 bone and mineral metabolism serum and urine results. In brackets, under each single biochemical parameter, the reference ranges of each are reported. In bold letters and gray background are reported results out of range.

II-1 Tw1	S-BSAP (5.0–31.0 mg/L)	S-CREAT (0.4–1.5 mg/dL)	S-Ca (8.1–10.4 mg/dL)	S-PO (2.5–5.0 mg/dL)	S-Mg (1.6–2.6 mg/dL)	P-OC (6–42 mg/L)	S-PTH (13–64 ng/L)	S-CTX (<540 ng/mL)	25OHD (≥30 ng/mL)
**November 2011**	14.4	0.6	9.1	4.2	1.9	**47**	52.9	**659**	**13**
**July 2013**	16.9	-	9.2	3.5	-	**44**	43.7	**739**	51.3
**March 2015**	21.0	0.6	9.4	3.4	2.0	37	46.9	**607**	**22.4**
**October 2019**	22.4	0.55	9.5	3.0	1.7	**49**	24.6	399	43.5
**November 2020**	23.6	0.66	9.7	3.1	2.1	42	52.7	474	39.7
	**24h U-Ca** **(100–300 mg)**		**24h U-CREAT.** **(600–1600 mg)**		**24h U-PO** **(200–1500 mg)**	**24 U-Mg** **(10–150 mg)**	**U-NTX** **(25–49 BCE */mmol CREAT.)**		**24 h** **DIURESIS mL**
**November 2011**	152		638		458	53	43		750
**November 2020**	168		745		553	82	**56**		1530
**II-2 Tw2**	**S-BSAP** **(5.0–31.0 mg/L)**	**S-CREAT.** **(0.4–1.5 mg/dL)**	**S-Ca** **(8.1–10.4 mg/dL)**	**S-PO** **(2.5–5.0 mg/dL)**	**S-Mg** **(1.6–2.6 mg/dL)**	**P-OC** **(6–42 mg/L)**	**S-PTH** **(13–64 ng/L)**	**S-CTX** **(<540 ng/mL)**	**25OHD** **(≥30 ng/mL)**
**October 2011**	16.4	0.7	9.1	3.4	1.8	38	**72.9**	**558**	**14**
**October 2018**	-	-	9.7	-	-	-	29.1	-	75.4
**12 November 2018**	28.5	0.65	9.1	-	1.7	-	-	-	-
**20 November 2018**	-	0.56	9.2	-	1.7	-	-	-	-
**28 November 2018**	-	0.72	9.3	-	1.68	-	-	-	-
**3 December 2018**	-	0.64	9.6	-	1.80	-	-	-	-
**13 December 2018**	-	0.69	9.4	-	1.94	-	-	-	-
**20 December 2018**	**40.1**	0.66	9.5	4.2	2.0	**192**	59.9	**1596**	34
**June 2019**	-	0.60	9.2	3.9	2.1	**106**	**86.2**	-	38.3
**October 2019**	**33.2**	0.67	9.6	5.0	1.7	30	56.4	-	54.8
**June 2020**	-	0.63	8.9	3.1	-	34	**76.2**	434	48.0
**December 2020**	28.6	0.76	9.2	4.0	1.9	24	**76.8**	-	58.1
	**24h U-Ca** **(100–300 mg)**		**24h U-CREAT.** **(600–1600 mg)**		**24h U-PO** **(200–1500 mg)**	**24 U-Mg** **(10–150 mg)**	**U-NTX** **(25–49 BCE */mmol CREAT.)**		**24 h** **DIURESIS mL**
**October 2011**	144		984		576	57	**65**		600
**June 2019**	165		888		603	62	**78**		1500
**December 2020**	203		786		597	71	**86**		1250

* BCE = bone collagen equivalents.

**Table 3 ijms-22-08331-t003:** Pharmacological therapy in place up to date in II-1 Tw1 and II-2 Tw2.

**II-1 Tw1**.
Calcifediol, 4 drops/day
Etanercept 50 mg, 1 subcutaneous vial/week
Acetylsalicylic acid, 100 mg/day
Levothyroxine, 50 mcg/day
Acyclovir 400 mg, twice/day
Fluconazole 200 mg, 2 tabs/day
Subcutaneous Immunoglobulin G-CSF, 30 mcg/mL, 30 mcg/day
**II-2 Tw2**.
Calcifediol 4 drops/day
Flurazepam 15 mg/day
Quietapine 25 mg/day
Estradiol hemihydrate patch twice a week and dihydrogesterone, derivatives of pregnadiene, 10 mg, 1 tab/day in the last 14 days of each 28-day cycle, sequentially;
Levothyroxine 75 mcg day for 5 days and 50 mcg/day for 2 days per week.

## Data Availability

Authors ensure that data shared are in accordance with consent provided by participants on the use of confidential data. Data sharing not applicable No new data were created or analyzed in this study. Data sharing is not applicable to this article.

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
