# Peer review of "A Novel Germline Mutation of ADA2 Gene in Two “Discordant” Homozygous Female Twins Affected by Adenosine Deaminase 2 Deficiency: Description of the Bone-Related Phenotype"

_ijms, 2021, doi:10.3390/ijms22158331_

Round 1
Reviewer 1 Report
The Authors present a case study of two 27 years-old DADA2 homozygote female twins exhibiting bone mineral density and bone turnover rate abnormalities over the years of their clinical follow-up.
This is a rare syndrome in which, in particular, aspects concerning the health of the skeleton are poorly understood.
The manuscript is very interesting and addresses for the first time specific issues relating to the ongoing bone metabolism of DADA2.
The Authors, after reviewing the pathophysiology of Ada2 enzymatic protein, discuss its potential role in bone health including the interactions with specific bone cell receptors.
The discussion is interesting and well argued, however the complexity of the clinical picture and polypharmacy over the years can themselves cause bone mineral density and bone turnover rate abnormalities. The above makes it very difficult to draw conclusions on the primary role of enzymatic deficiency (DADA2) on bone metabolism.
I therefore recommend that you better highlight these limits.
Author Response
The authors are grateful to the reviewer for the nice words expressed on this paper and for the suggestions provided which will help improve it. All changes/modifications introduced are highlighted in yellow in the text and bibliography. The letter Q identifies the question posed by the reviewer, while the letter A identifies the relative answer provided by the authors.
Q1/ The discussion is interesting and well argued, however the complexity of the clinical picture and polypharmacy over the years can themselves cause bone mineral density and bone turnover rate abnormalities. The above makes it very difficult to draw conclusions on the primary role of enzymatic deficiency (DADA2) on bone metabolism.
I therefore recommend that you better highlight these limits.
A1. A final paragraph titled 6. Limitations and Conclusions has now been introduced just to accommodate this suggestion.
Reviewer 2 Report
In this case report, authors describe twins affected by a rare genetic immune disorder, adenosine deaminase 2 deficiency (DADA2), with particular emphasis on the bone phenotype. Limitations are adequately discussed.
MAJOR COMMENTS:
1/ Some further grammar/language editing is required e.g.: "growth failure as also other skeletal abnormalities" should be "as well as" (not "as also), or sentences like "due to vasculitis events occurrences" or "All the clinical-instrumental and biochemical parameters to investigate bone health 174 in our cases are here specifically treated." (what does "treated" mean here? Probably a translation problem)
2/ Consanguinity of the parents or not, should be reported (probably it is), as well as ethnicity. Also, are the twins monozygotic or dizygotic?
3/ Why was informed consent necessary for a clinical work-up? Please specify the date on which the Ethical Committee gave approval, and a reference number if possible.
4/ The authors reiterate an old myth that trabecular bone reflects 80% of bone turnover. Reality is much more complicated. Please revise.
5/ The authors describe homozygous twins in their abstract but report the presence of a heterozygous mutation, how can they then assert homozygosity? Surely their conclusion that residual enzyme activity was very low is important and supports a complete deficiency, but as the authors discuss themselves, there are several possible mechanisms (including a dominant-negative effect). Thus, the conclusion of "homozygosity" should be removed.
MINOR COMMENTS:
6/ There are some references formatted as "PubMed:8452534", these need to be removed
7/ Do not abbreviate single words like osteoclasts (OCL) or osteoblasts (OBL), and CERTAINLY avoid hybrid abbreviations like OBLgenesis for osteoblastogenesis.
8/ PAN is more commonly expanded as polyarteritis nodosA (instead of nodosUM)
9/ There is a reference to a cancerresearchuk.org website, text should not be copied literally even with reference to the source, suggest rewriting the text and removing the intext link.
10/ RANKL is "Receptor acivatOR" (not activatING)
Author Response
Reply to reviewer 2
The authors are grateful to the reviewer for the constructive criticisms made to our paper, right and correctly expressed, which will help improve it. All changes/modifications introduced are highlighted in yellow in the text and bibliography. The letter Q identifies the question posed by the reviewer, while the letter A identifies the relative answer provided by the authors.
MAJOR COMMENTS:
Q1/ Some further grammar/language editing is required e.g.: "growth failure as also other skeletal abnormalities" should be "as well as" (not "as also), or sentences like "due to vasculitis events occurrences" or "All the clinical-instrumental and biochemical parameters to investigate bone health in our cases are here specifically treated." (what does "treated" mean here? Probably a translation problem);
A1. We thank the reviewer for his/her careful reading and clarifications made that certainly contribute to improving our paper. All the suggestions have been now incorporated within the text (highlighted in yellow). In particular, the term treated has been modified as “reported”;
Q2/ Consanguinity of the parents or not, should be reported (probably it is), as well as ethnicity. Also, are the twins monozygotic or dizygotic?
A2. Good point. Now, we introduced the following sentence (highlighted in yellow) “Information was collected, as detailed as possible, aimed at ascertaining/excluding possible consanguinity of the parents, going back to the great-grandparents, from which no data supporting the consanguinity of the parents emerged, although this procedure does not allow it to be absolutely excluded.” The twins are monozygotic.
Q3/ Why was informed consent necessary for a clinical work-up? Please specify the date on which the Ethical Committee gave approval, and a reference number if possible.
A3. We appreciate the observation made by reviewer. Now, we have introduced the following sentence (highlighted in yellow): “Both twins gave their informed consent, as an internal policy of the institution approved by the Internal Board, for the treatment of personal data, for the clinical-biochemical, genetic, ionizing radiation, procedures and neridronate treatment (II-2 Tw2). Signed in-formed consent was also obtained from the parents for the genetic analysis and data management.” As rightly doubted by the reviewer, no ethical committee approval was necessary for the clinical work-up. We sincerely apologize for this silly inattention due to a "typical cut and paste error"
Q4/ The authors reiterate an old myth that trabecular bone reflects 80% of bone turnover. Reality is much more complicated. Please revise.
A4. We completely agree with the reviewer regarding this clarification, this concept has now been removed from the text. Being a complex and complicated concept, we do not consider it adequate to treat it in this case report, nor an extreme simplification, as we tried to do, which led to propose a previous myth, now no longer corresponding to the strict reality.
Q5/ The authors describe homozygous twins in their abstract but report the presence of a heterozygous mutation, how can they then assert homozygosity? Surely their conclusion that residual enzyme activity was very low is important and supports a complete deficiency, but as the authors discuss themselves, there are several possible mechanisms (including a dominant-negative effect). Thus, the conclusion of "homozygosity" should be removed.
A5. We thank the reviewer for his/her brilliant consideration and removed the term homozygous as suggested. However, we considered it correct to insert the term monozygotic as it is a question of two identical twins individuals deriving from a single zygote (the egg fertilized by the sperm).
MINOR COMMENTS:
Q6/ There are some references formatted as "PubMed:8452534", these need to be removed
A6. We apologize for this typo. Both PMIDs have been deleted and the correct references inserted, in the text and in the bibliography.
Q7/ Do not abbreviate single words like osteoclasts (OCL) or osteoblasts (OBL), and CERTAINLY avoid hybrid abbreviations like OBLgenesis for osteoblastogenesis.
A7. All corrections / changes indicated by the reviewer have been made.
Q8/ PAN is more commonly expanded as polyarteritis nodosA (instead of nodosUM)
A8. In perfect agreement with the reviewer, the text has been changed as required.
Q9/ There is a reference to a cancerresearchuk.org website, text should not be copied literally even with reference to the source, suggest rewriting the text and removing the intext link.
A9. We have followed the suggestions given with the required changes.
Q10/ RANKL is "Receptor acivatOR" (not activatING)
A10. We also apologize for this typo. Now this problem has been solved